# Modified Cu-Sn Catalysts Enhance CO2RR Towards Syngas Generation

**DOI:** 10.3390/ma18174070

**Published:** 2025-08-30

**Authors:** Daniel Herranz, Antonio Maroto, Martina Rodriguez, Juan Ramón Avilés Moreno, Pilar Ocón

**Affiliations:** Departamento de Química Física Aplicada, Universidad Autónoma de Madrid, C/ Francisco Tomás y Valiente 7, 28049 Madrid, Spain; daniel.herranz@uam.es (D.H.); antonio.maroto@estudiante.uam.es (A.M.); martina.rodriguez@estudiante.uam.es (M.R.); pilar.ocon@uam.es (P.O.)

**Keywords:** CO_2_ reduction reaction, electrocatalysis, electrodeposition, Cu, Sn, bimetallic catalyst

## Abstract

The electrochemical reduction in CO_2_ (CO2RR) to syngas and value-added hydrocarbons offers a promising route for sustainable CO_2_ utilization. This work develops tuneable Cu-Sn bimetallic catalysts via electrodeposition, optimized for CO2RR in a zero-gap flow cell fed with CO_2_-saturated KHCO_3_ solution, a configuration closer to industrial scalability than conventional H-cells. By varying electrodeposition parameters (pH, surfactant DTAB, and metal precursors), we engineered catalysts with distinct selectivity profiles: Cu-Sn(B), modified with DTAB, achieved 50% Faradaic efficiency (FE) to CO at −2.2 V and −50 mA·cm^−2^, outperforming Ag-based systems that require higher overpotentials. Meanwhile, Cu-Sn(A) favoured C_2_H_4_ (35% FE at −100 mA·cm^−2^), and Cu-Sn(C) shifted selectivity to CH_4_ (26% FE), demonstrating product tunability. The catalysts’ performance stems from synergistic Cu-Sn interactions and DTAB-induced morphological control, as revealed by SEM/EDX and electrochemical analysis. Notably, all systems operated at lower voltages than literature benchmarks while maintaining moderate CO_2_ utilization (32–49% outlet). This study highlights the potential of electrodeposited Cu-Sn catalysts for energy-efficient CO2RR, bridging the gap between fundamental research and industrial application in syngas and hydrocarbon production.

## 1. Introduction

The excessive emission of carbon dioxide (CO_2_) from fossil fuel combustion, industrial processes, and land-use changes represents the primary driver of global climate change, accounting for over 50% of observed temperature increases [1]. Transitioning to a carbon-neutral energy cycle is imperative for sustainability, with current strategies focusing on: (i) CO_2_ emission reduction, (ii) carbon capture and storage (CCS), and (iii) carbon capture and utilization (CCU) [2]. Among these, CCU stands out by valorizing CO_2_ as a feedstock for high-value products, which can be achieved through electrochemical CO_2_ reduction (CO2RR) powered by renewable electricity [3].

CO2RR to carbon monoxide (CO) is particularly promising due to its techno-economic viability [4] and utility in syngas production (CO:H_2_ ≈ 1:2 ratio), a critical precursor for Fischer–Tropsch (FT) synthesis and methanol production [5,6]. Current CO2RR systems employ three main configurations: H-cells, flow cells, and membrane electrode assembly (MEA) cells. While H-cells and flow cells enable reference electrode integration, their liquid electrolyte layers incur high ohmic losses. In contrast, MEA cells minimize resistance via zero-gap designs [7,8].

Many researchers utilize gaseous CO_2_ feed to enhance CO selectivity [9,10,11], but recent advances highlight carbonate/bicarbonate solutions as alternatives. These systems deliver CO_2_ via two pathways: (i) dissolved aqueous CO_2_ stabilized by KHCO_3_ [7], or (ii) in situ CO_2_ generation at the cathode-membrane interface using protons from bipolar membranes [12,13]—albeit at high overpotentials [14,15]. Liquid-phase CO_2_ feeding improves utilization rates [16] and integrates seamlessly with alkaline carbon capture [17].

The reactions are:

Cathode:(1)$$CO2RR\;to\;CO\left(acidic\right):\;\;\;CO_{2}+2H^{+}+2e^{-}\to CO+H_{2}O$$(2)$$CO2RR\;to\;CO\left(alkaline\right):\;\;\;CO_{2}+H_{2}O+2e^{-}\to CO+2OH^{-}$$(3)$$HER\;\left(acidic\right):\;\;\;2H^{+}+2e^{-}\to H_{2}$$(4)$$HER\;\left(alkaline\right):\;\;\;2H_{2}O+2e^{-}\to H_{2}+2OH^{-}$$

Anode:(5)$$OER\;\left(acidic\right):\;\;\;2H_{2}O\to O_{2}+4H^{+}+4e^{-}$$(6)$$OER\;\left(alkaline\right):\;\;\;4OH^{-}\to O_{2}+{2H}_{2}O+4e^{-}$$

Catalyst development for CO2RR spans coprecipitation [18,19], chemical etching [20,21], hydrothermal synthesis [22,23], and electrodeposition [24,25]. Electrodeposition offers distinct advantages: environmental benignity, cost-effectiveness (using abundant metal salts), binder-free fabrication, and precise control over morphology for enhanced stability and mass transport.

Electrosynthesized catalysts for CO2RR primarily fall into two categories: single-metal and bimetallic systems. Single-metal variants exhibit distinct product selectivity, with Pd, Sn, and Bi-based catalysts typically yielding formic acid (HCOOH) [26,27,28], while Zn, Ag, and Au catalysts predominantly generate CO [29,30,31,32]. Copper electrodes, when surface-modified, demonstrate remarkable versatility, capable of producing HCOOH, CO, CH_4_, and C_2_₊ products such as C_2_H_4_ [33,34,35,36].

The emergence of bimetallic catalysts has marked a significant advancement in CO2RR research, taking advantage of the synergistic electronic and chemical interactions between constituent metals to enhance catalytic performance. Despite their demonstrated potential, key challenges remain in precisely controlling their composition, structure, and mechanistic understanding. Among various metal combinations, Cu-Sn systems have garnered particular attention due to their cost-effectiveness as non-precious metals and promising catalytic performance.

Notable examples include the work of Wang et al., who achieved 84% CO faradaic efficiency using Cd-Cu catalysts prepared through electrodeposition and operated at −1.0 V vs. RHE [37]. Similarly, Li et al. developed Cu-Sn alloys that delivered exceptional 91.38% FE toward HCOOH production at −0.8 V vs. RHE, while systematically investigating metal ratio effects on catalyst morphology [38]. These promising results have spurred further investigation into Cu-Sn systems [39,40,41,42], though most studies to date have been limited to conventional H-cell configurations with continuous gaseous CO_2_ feed to the cathode.

The electrochemical reduction in CO_2_ to CO represents the most straightforward reaction pathway, involving a two-electron and two-proton transfer mechanism at the catalyst surface. Extensive studies have identified gold and silver as the most selective catalysts for this conversion [43,44], prompting significant research into their surface reaction mechanisms. These noble metals demonstrate exceptional performance for syngas generation [45,46,47,48,49], with gold exhibiting superior selectivity despite being 50–70 times more costly than silver [50]. This substantial price differential has driven considerable efforts to optimize silver-based catalysts for efficient CO production. The abundance of studies based on silver for CO production makes this family of catalysts especially interesting as a comparison point for non-noble metal, inexpensive catalysts that can effectively and selectively produce CO.

For more complex product distributions, copper-based catalysts show remarkable potential in generating C_1_ and C_2_ organic compounds (hydrocarbons and alcohols) at practical current densities, as demonstrated by Hori et al. [43,44] and Azuma et al. [49]. Alternative approaches include alloying Cu with other elements to uniquely manipulate the reactivity and selectivity by tuning the local electronic structure and modulating the adsorption strength of the reaction intermediates, thus changing the catalytic properties of the catalyst [51]. For example, the study of Shang et al. demonstrated that a low-entropy state Cu_3_Sn catalyst allows a high Faradaic efficiency of 64% for ethanol production, distinctively from the high-entropy state Cu_6_Sn_5_ catalyst with the main selectivity toward producing formate [52].

A particularly innovative approach by Stojkovik et al. [42] aligns with circular economy principles, utilizing recycled industrial Cu-Sn bronze waste to create effective CO2RR catalysts with competitive CO production efficiencies. Another interesting approach for optimizing CO2RR has been made in the study of Wuttig et al. [53], where they show how CO2RR catalysts can be poisoned by trace metal ion impurities present in high purity electrolytes and how it can be avoided by the use complexing agents, finding an improved performance with ethylenediaminetetraacetic acid (EDTA) and highlighting the relevance of the fine control of CO2RR conditions.

This work advances Cu-Sn catalysts for CO2RR in zero-gap flow cells fed with CO_2_-saturated KHCO_3_ solutions—a configuration bridging lab-scale research and industrial applicability. In our previous study, we tackled different combinations of Cu and Sn for CO2RR, highlighting the synergistic behaviour between the two metals [54]. Here, we systematically investigate how electrodeposition parameters (metal precursors, pH, and surfactant modification) govern catalytic performance across a practical current density range (−25 to −100 mA·cm^−2^) and flow rates (20–120 mL·min^−1^). By correlating synthesis conditions with CO_2_ conversion efficiency, product selectivity (CO, CH_4_, C_2_H_4_), and stability, we identify optimal formulations for syngas and hydrocarbon production while minimizing energy penalties.

## 2. Materials and Methods

The following reagents were directly used: KOH (85%) from Labbox (Barcelona, Spain), KHCO_3_ (99%) and ethylenediaminetetraacetic acid (EDTA) from Scharlab (Barcelona, Spain), Fumasep FAA-3-PE-30 membranes from FuMa-Tech (Bietigheim-Bissingen, Germany), nickel foam (99.99%) from Nanografi Nano Technology (Ankara, Turkey), GDL carbon cloth with carbon MPL and treated with PTFE from Fuel Cell Store (Bryan, TX, USA), CuCl_2_·2 H_2_O (99.99%) and CuSO_4_·5 H_2_O (99.99%) from Panreac (Barcelona, Spain), SnCl_2_·2 H_2_O (99.99%) from Quality Chemicals (Esparreguera, Spain) and gaseous compressed CO_2_ (99.7%) from AirLiquide (Madrid, Spain).

The electrolyte used for the electrodeposition process was a solution of 0.01 M EDTA and 0.05 M CuCl_2_ or CuSO_4_ as Cu precursors or 0.05 M SnCl_2_ as Sn precursor. pH regulation was achieved with H_2_SO_4_ or HCl when SO_4_^2−^ or Cl^−^ salts were used, respectively. pH values were selected according to the Pourbaix diagrams of Cu and Sn to present stable ions in solution that could be reduced for electrodeposition, but avoiding too low pHs and the corresponding extensive HER. DTAB was also added to some of the solutions as described in the results section. The catalyst fabrication was conducted in a custom three-electrode electrochemical cell configuration. A carbon cloth gas diffusion layer (GDL) served as the working electrode, mechanically pressed against a stainless steel current collector at the cell base while immersed in electrolyte. The electrochemical system was completed with a graphite rod counter electrode and a saturated Ag/AgCl (KCl) reference electrode. Potential/current control was maintained using an Autolab PGSTAT302N potentiostat/galvanostat from Metrohm (Madrid, Spain), with deposition performed via either chronopotentiometric or chronoamperometric methods as detailed in the Results section.

For bimetallic Cu-Sn catalysts, a sequential deposition approach was employed: first, electrodepositing Cu onto the carbon cloth substrate, followed by Sn deposition onto the Cu underlayer. Post-deposition, all catalysts underwent rigorous washing with distilled water to eliminate residual electrolytes, followed by overnight drying at 60 °C. Catalyst loading was determined gravimetrically by comparing the mass of pristine and metal-deposited carbon cloth, yielding typical loadings of 2–6 mg·cm^−2^.

Catalyst morphology was examined using an Olympus BX41 optical microscope from Evident (Barcelona, Spain) equipped with four objective lenses providing 5×, 10×, 20×, and 50× magnification capabilities. For a deeper analysis of the samples’ morphology and composition, surface and morphology images were captured using a Hitachi S–3000N SEM (Tokio, Japan) coupled with an EDX XFlash^®^ 6130 Bruker detector (Berlin, Germany) for semi-quantitative chemical composition analyses. Internal standard quantitative analyses were used for EDX quantification. The equipment operated under high vacuum conditions with an accelerating voltage of 20 keV and a working distance of 15.0 mm. The main elements detected are presented in the corresponding figures, apart from carbon, coming from the carbon cloth, and trace elements that may come from ambient contamination.

The CO_2_ electrolysis experiments employed a commercial electrochemical flow cell (ElectroChem Inc., Raynham, MA, USA) featuring gold-coated current collector plates compressing two graphite flow plates. A precisely defined 4 cm^2^ active area (2 cm × 2 cm) was created using silicon and PTFE gaskets that simultaneously prevented electrolyte/gas leakage. The cell incorporated a Fumasep FAA-3-PE-30 membrane (3 cm × 3 cm) to separate the anodic and cathodic chambers, with nickel foam and electrodeposited carbon cloth serving as anode and cathode catalysts, respectively, of the two-electrode configuration cell.

A dual-channel peristaltic pump (Dinko Instruments D-25V) circulated the electrolytes: anolyte with 0.5 M KOH aqueous solution and catholyte with 0.5 M KHCO_3_ + 0.01 M EDTA (to chelate metal impurities [53]). The catholyte was continuously saturated with CO_2_ via syringe injection to maintain dissolved CO_2_ concentration.

The experiments were performed using an Autolab PGSTAT302N potentiostat/galvanostat from Metrohm (Madrid, Spain), applying current densities from −25 to −100 mA·cm^−2^ with electrolyte flow rates of 20–120 mL·min^−1^. Each condition was maintained for ≥5 min to accumulate sufficient gaseous products for analysis and the different conditions were tested consecutively with the catalyst assembled in the flow cell.

Gaseous products were sampled using a 5 mL SGE gas-tight syringe and analyzed by gas chromatography (Varian 3900 with Carboxen-1006 PLOT column) from Análisis Vínicos (Tomelloso, Spain) coupled to mass spectrometry (Pfeiffer Vacuum Hi-Cube) from Tecnovac (Alcobendas (Madrid), Spain). The GC (Gas Chromatography) method employed argon carrier gas with a temperature ramp from 35 °C to 245 °C at 30 °C·min^−1^ for optimal compound separation and detection. Comprehensive monitoring of both catholyte and anolyte solutions was performed using GC-FID/TCD (Gas Chromatography coupled to Flame Ionization/Thermal Conductivity Detectors) techniques employing a Thermo TRACE™ GC Ultra Gas Chromatograph from Análisis Vínicos (Tomelloso, Spain). No organic compounds from CO2RR were detected in either liquid phase, confirming all reported reaction products were exclusively gaseous in nature.

The gas composition analysis followed a two-step normalization procedure: first, CO_2_ outlet concentration was expressed as a percentage of total gas volume, and then CO2RR and HER products (CO, CH_4_, C_2_H_4,_ and H_2_) were normalized to 100% for Faradaic efficiency calculations. This normalization approach results in the sum of reaction products and CO_2_ percentages visually exceeding 100%.

The efficiency for each product i was determined by:$${FE}_{i}=\frac{z_{i}\cdot F\cdot x_{i}\cdot n}{Q}$$
where $z_{i}$ is the number of electrons transferred per molecule (2 for CO, 2 for H_2_, 8 for CH_4,_ and 12 for C_2_H_4_), F is the Faraday constant (96,485.33 C·mol^−1^), $x_{i}$ is the molar fraction of the product i, n is the total number of mols in the products, and Q is the total charge passed.

## 3. Results and Discussion

### 3.1. Performance of Catalysts Synthesized Under Constant Current Electrodeposition

One of the methods tested for catalyst (Cu-Sn) synthesis was the electrodeposition by chronopotentiometry (CP). The first metal to be electrodeposited is Cu, and the initial conditions were an aqueous solution of 50 mM CuSO_4_ acidified with H_2_SO_4_ until pH 2 was reached. First, a CV of the Cu solution was carried out with the carbon cloth as the working electrode; the results are included in Appendix A.

The current density selected was 7 mA·cm^−2^ for Cu electrodeposition and two samples were prepared, one with the previously mentioned solution, Cu(SO_4_), and another with the same conditions but using CuCl_2_ instead of CuSO_4_, in order to determine which one works better. In both samples, Cu was electrodeposited in two consecutive CPs of 20 min each. The Cu catalysts synthesized were tested for CO2RR at −25 and −50 mA·cm^−2^ current densities in the MEA-flow cell, using electrolyte flow rates of 3 to 80 mL·min^−1^. The results are shown in Figure 1 and Appendix A.

The Cu(SO_4_) catalyst shows lower FE to CO compared to CuCl_2_, but it shows greater FE to CH_4_ and C_2_H_4_. The relatively high abundance of CH_4_ and C_2_H_4_ was expected since Cu is known to transform CO_2_ efficiently to C_1_ and C_2_ products compared with transformation to CO [55]. There are several interesting tendencies: CO_2_ outlet always increases with increasing electrolyte flow rate, FEs to CO are better at −25 mA·cm^−2^, but FE to CH_4_ and C_2_H_4_ are better at −50 mA·cm^−2^, and C_2_H_4_ always shows a maximum at 20 mL·min^−1^ electrolyte flow rates. The observed change in FEs to specific products between the two catalysts reveals the importance of the metal precursor salt used in the electrodeposition process, showing in this case that CuCl_2_ is better for CO production when only Cu is electrodeposited. The affinity of the counter-ion to the carbon cloth, higher for SO_4_^2−^ than Cl^−^ [56], is probably responsible for the different superficial morphology of the electrodeposited Cu, which relates to the adsorption energy of CO_2_, intermediates, and final products. The higher amount of organic products at increased reduction current and voltage has been previously observed by other authors, like the study of Gao et al. [57], and the actual explanation proposes that it is produced by the faster transformation to them before the molecule is desorbed from the catalyst surface, due to the adequate adsorption strength of the intermediates over Cu. Cu(SO_4_) catalyst obtains a high 29% FE to C_2_H_4_, which is not the main focus of this work, but shows the relevant potential of this catalyst for ethylene production. The increase in outlet CO_2_ with increasing electrolyte flow rate was also observed in our previous study and has also been reported by other authors [58], relating it with the facilitated desorption of the CO_2_ molecule in a probably more turbulent environment before it is able to react.

SEM/EDX analysis was performed on these catalysts in order to study in more detail the surface morphology and composition (Figure 2).

The catalyst Cu(SO_4_) shows a relevant change before and after being used in CO2RR from the SEM images, losing most of the metal deposited over the carbon cloth after the CO2RR process and showing some Nafion^®^ particles with high F content coming from the carbon cloth. Probably, as a consequence of the continuous flow of electrolyte, the catalysts are detaching, as previously observed in other works [54,59]. The Cu structure present before CO2RR reveals Cu dendrites with high Cu atomic % and almost no O. After CO2RR, most of the electrodeposited metal has been washed away. The Cu(Cl2) catalyst shows a different behaviour: first, the remaining amount of catalyst after CO2RR on the carbon cloth surface is clearly higher, and second, the metal structures present a different shape. The better attachment of Cu structures may be related to the lower affinity of Cl^−^ ions for chemisorption to the carbon cloth compared with the SO_4_^2−^ ions. That chemisorption of SO_4_^2−^ could prevent a more efficient adhesion of the metallic structures to the supporting surface. The presence of Cl^−^ ions in solution is probably related to the formation of pure Cu polyhedral cubes instead of dendrites. They present smaller formations partially on top with some oxygen content. All these structures show a clear transformation after CO2RR, developing concentric needles composed of Cu oxide or Cu hydroxide. Based on the study of Qui et al. [60], these structures were identified as Cu(OH)_2_. Cu(OH)_2_ is usually considered to be unstable in solution, but according to Cudennec et al. [61], it can be stable when the (OH)^−^ concentration is low. This is probably the case, since the pH of the electrolyte is between 7 and 8, and the (OH)^−^ concentration increases locally near the Cu surface due to HER, but the excess is rapidly removed by the electrolyte flow.

In this work, the main CO2RR product focuses on CO for the syngas production; therefore, the electrodeposition at constant current continued using CuCl_2_ solution. Based on that, two catalysts Cu-Sn, were prepared and the effect of the initial pH during CuCl_2_ electrodeposition was compared. The first one was prepared by adding HCl to acidify the CuCl_2_ solution to pH 2 and the second one leaving the solution at pH 4.5. After Cu electrodeposition, Sn was electrodeposited from 50 mM SnCl_2_ solution at pH 4.5 in both cases, obtaining Cu-Sn bimetallic catalysts. The first one is named Cu-Sn(2) and the second Cu-Sn(4.5). The performance of these catalysts for CO2RR is presented in Figure 3 and Appendix A.

Both catalysts show the soft tendency of increasing CO_2_ outlet with the higher electrolyte flow rate previously mentioned, but in other aspects, their behaviour is different. On one hand, Cu-Sn(2) catalyst obtains low FE values for CO, CH_4,_ and C_2_H_4_. On the other hand, Cu-Sn(4.5) presents high FE, specifically for CO as expected, but also to C_2_H_4_ and CH_4_, reaching FE_CO_ 50% at −50 mA·cm^−2^ and 120 mL·min^−1^ or FE_CH4_ 9.2% at the same current density but 40 mL·min^−1^. The improved production of CO when the metal is electrodeposited at higher pH may arise from an optimized adhesion or structure under minimized HER conditions. In this very good performance catalyst, CO production rises with increasing electrolyte flow rate, similar to previous Cu catalysts, but the increase in current density also enhances FE to CO, opposite to those. It can therefore be concluded that the improvement in performance is due to the combined effect of pH and Sn addition during catalyst synthesis.

Pictures of the catalysts’ surface as synthesized are presented in Appendix A. From a macroscopic view, catalyst Cu-Sn(2) shows a highly heterogeneous Sn distribution over Cu, with broad red Cu areas visible under the grey Sn, whereas Cu-Sn(4.5) has Sn homogeneously distributed. However, a closer look with the optical microscope shows that Sn does not have a completely uniform distribution and Cu is visible under it. Since Cu-Sn(4.5) catalyst obtained the best results, it was studied in more detail by SEM/EDX analysis, and the results are depicted in Figure 4.

It is relevant that Cu-Sn(4.5) catalyst overall coverage is maintained better than Cu(Cl2) catalyst and much more than Cu(SO_4_), which is indicative of its better stability. Pre-CO2RR analysis reveals a mixture of agglomerates without specific structures identified for the different metals. The composition of these agglomerates is about 40% Cu and 20% Sn with low amounts of O. The higher presence of Cu, on which Sn was electrodeposited, confirms the visual observation in Appendix A and the fact that Sn was not homogeneously electrodeposited. It may also come from a lower amount of electrodeposited Sn and the good penetration of the EDX analysis on the sample surface, reaching the Cu underneath areas covered by Sn. After CO2RR, relevant changes are identified: many of the agglomerates look partially destroyed with holes in the structures, which may arise from the detachment of part of the catalyst by the electrolyte flow. New structures are also observed, showing a star-like morphology that, based on the EDX analysis, is made mainly of CuO_2_. However, most of the surface is still covered by the Cu-Sn agglomerates with a composition like the one previous to CO2RR.

Overall, the Cu-Sn(4.5) catalyst presents very promising results with high FE to CO and stability, but the homogeneity of the electrodeposition needs to be improved.

### 3.2. Performance of Catalysts Synthesized Under Constant Voltage Electrodeposition

Based on the previous results and the importance of good homogeneity and adhesion of the electrodeposited metals, the study of catalyst optimization is approached by chronoamperometry. It starts with a 50 mM Cu SO_4_ solution at pH 2, similar to the previous one. To improve the homogeneity of the catalysts, the use of the DTAB additive was investigated. The CVs with and without 5 mM DTAB in the CuSO_4_ solution are presented in Figure 5.

The presence of DTAB interacts with the surface of the carbon cloth attaching to it the polar end and presenting the apolar edge towards the solution [62]. This effect diminishes the interaction of the species in solution with the polarized surface, which is probably the reason for the lower reduction and oxidation currents of Cu as well as the lower HER intensity at relatively high negative potentials. This can lead to a more homogeneous electrodeposition due to a less turbulent environment at lower HER rates and more electrons used for the reduction in the metals instead of H_2_ formation. The lower reduction currents also contribute to better adhered and more homogeneous electrodeposition thanks to the slower process.

From the obtained CVs, several voltage values were selected to test short CAs and evaluate the currents produced and the homogeneity of the electrodeposited samples. Results are presented in Appendix A.

The current densities obtained are very close, but when the solution had DTAB in it, the current densities are slightly less negative, which is consistent with the previously explained effect of the additive. Visually, it was observed that when DTAB was present, relatively high negative voltage was needed to obtain good coverage of the carbon cloth. Therefore, the voltage used in the following experiments was higher (−0.9 V) when DTAB was used and lower (−0.6 V) when the additive was not employed.

The bimetallic catalysts were prepared by combining the test of using DTAB or not in Cu electrodeposition and the use of SnSO_4_ or SnCl_2_ (50 mM both) as precursor salts for the Sn electrodeposition (see Table 1). Prior to each experiment, a CV was performed and the optimal potential for the electrodeposition of CA was selected.

Based on the previous results, bimetallic catalysts were prepared to test the CO2RR performance in the MEA-flow cell. The results are presented in Figure 6 and Appendix A.

When DTAB is only used in the Cu layer, catalyst Cu-Sn(A), good FE to CO is obtained at low current density, reaching its maximum at −25 mA·cm^−2^ with FE_CO_ 43%. This catalyst, similar to the others, shows an increasing formation of CH_4_ and C_2_H_4_ with more negative currents, reaching with this one the highest C_2_H_4_ FE of the catalysts tested in this study, with 35% at −100 mA·cm^−2^ and 80 mL·min^−1^. Regarding CH_4_ production, the best values are obtained by Cu-Sn(C) catalyst, with FE_CH4_ 26% at −100 mA·cm^−2^ and 80 mL·min^−1^, similar values to those previously obtained by catalyst Cu(SO_4_). Although the focus of this study is to improve CO production, the high efficiency shown by these catalysts for CH_4_ and C_2_H_4_ makes them attractive candidates for C_1_ and C_2_ added-value products generation. Finally, the two catalysts that used DTAB during the electrodeposition of Cu and Sn, Cu-Sn(B), and Cu-Sn(C), also obtain very good values of FE to CO, reaching their best performance at −50 mA·cm^−2^, opposed to Cu-Sn(A) at −25 mA·cm^−2^. That may be an indication of the influence of DTAB, allowing formation of structures that perform better at higher current densities. Among these catalysts, the best FE to CO was obtained by Cu-Sn(B), reaching 50% at −50 mA·cm^−2^ and 120 mL·min^−1^. This very good FE is similar to the one obtained with catalyst Cu-Sn(4.5) using the same current density and electrolyte flow rate; however, the potential during CO2RR with that catalyst was −2.46 V and using Cu-Sn(B), the potential is −2.17 V, which translates to a lower energy consumption and thus more efficient CO production. H_2_ production, as a competitive reaction, follows a trend opposite to the production of CO2RR products, which in most cases is opposite to CO production. These catalysts reach compositions of CO-H_2_ around 33–66%, in the range of syngas composition, so they could be highly desirable for industrial syngas production.

The SEM/EDX analysis of the catalysts before and after CO2RR is depicted in Figure 7 and Appendix A.

Cu-Sn(A) catalyst before CO2RR shows a homogeneous structure, mostly covered by Sn in flower-like shapes with some but not high O content, with small areas where some Cu is present with Sn in a diffuse shape. After CO2RR, the surface of the carbon cloth is still well covered by the catalyst, but most of the Sn has vanished, and the majority of the detected metal is Cu. A similar effect was observed by XPS in our previous work [54]. Sn still shows a flower-like structure quite similar to the one before CO2RR, but with a much higher content in Cu, and Cu has evolved to cover most of the surface with a foil-like structure with needles at the borders composed mainly of Cu with a high ratio of O, probably Cu oxides or hydroxides. Even if the shape is different from the concentrical needles of Cu(Cl2) catalyst or the star-like needles of Cu-Sn(4.5) catalyst, the composition is quite similar; they show an expanding shape and in all cases they appear after the catalyst has been used for CO2RR, suggesting they have been formed during the process.

Looking at Cu-Sn(B) catalyst, before CO2RR, the carbon cloth surface is covered with a heterogeneous mixture of both metals, with clearly separated areas for one and the other. After CO2RR, most of the Sn is lost and three main areas can be distinguished, one with still some Sn mixed with the Cu, one mainly composed of Cu with some O content, and one also of Cu but with high O content and flag-like shapes.

For the Cu-Sn(C) catalyst, the aspect of the sample before CO2RR is quite a homogeneous mixture of Cu and Sn, as the mapping shows, especially in comparison with Cu-Sn(B) (Appendix A), with mainly amorphous structures made of aggregates of small particles. The amount of Cu and Sn varies between different areas, but there are no specific shapes that can be identified as made of one of the metals clearly. The high O content may indicate that a relevant percentage of the metal is in oxide or hydroxide form. After CO2RR, the situation is quite similar to the previous catalysts, losing most of the Sn on the carbon cloth surface. Some agglomerate particles made of Cu and Sn with high O content are visible as well as new shapes made of Cu and O. These ones are made of needles and look between the concentrical needles of Cu(SO_4_) and the star-like shapes of Cu-Sn(4.5) after CO2RR, thus most probably they are formed in the same manner.

The above results show that the addition of DTAB to the SnCl_2_ solution does not improve the homogeneity of the Sn layer, but it makes the opposite. They also reveal the formation of expanding Cu-made structures, probably of Cu(OH)_2_, during CO2RR, which will be further investigated in the future to elucidate their role in the production of products. Finally, some of the catalysts with low homogeneity have obtained the best FEs, showing that it is probably the optimization of the mixture and areas of interaction of the two metals the key to improving the performance of these catalysts.

A critical evaluation of current literature reveals that most CO2RR studies employ conventional H-cell configurations, where CO_2_ is introduced directly into the cathode chamber. In such systems, the reactant reaches the electrode surface through both dissolved CO_2_ and gas bubble transport, significantly enhancing CO_2_ availability while suppressing the competing hydrogen evolution reaction (HER). These systems typically report CO Faradaic efficiencies (FE) of 80–90%, though they frequently omit crucial CO_2_ utilization metrics.

The present flow-cell system, operating without direct CO_2_ bubbling, is closer to carbonate/bicarbonate-fed reactors (see Table 2). As comprehensively reviewed by Li and Shao [63], state-of-the-art bicarbonate systems achieve CO FE values of 82% [64] and 60% [58], but require substantially higher cell voltages (−3.4 V and −3.7 V, respectively). Notably, high-performance CO-selective catalysts in such configurations predominantly utilize silver, while Cu- or Sn-based alternatives tend to favor other products while still operating at elevated potentials [65,66].

Our group’s prior work with bipolar membranes [67,68] encountered similar voltage requirements, attributable to the additional overpotential needed for water splitting for in situ CO_2_ generation. While cation exchange membrane systems can achieve lower voltages (−2.2 V), their CO FE remains limited to 15% [69]. In contrast, our previous study [54] demonstrates superior performance, achieving 62% CO FE at just −2.0 V with 49% CO_2_ utilization.

For multi-product scenarios, Cu-Sn(A) achieves 35% FE to C_2_H_4_ at −100 mA·cm^−2^—a rare feat for bicarbonate-fed systems—while maintaining moderate CO_2_ utilization (35% outlet). Meanwhile, Cu-Sn(C) shifts selectivity toward CH_4_ (26% FE), demonstrating tunability via electrodeposition design. Both variants operate at −2.7 V, far below the −7.2 V required for comparable CH_4_ production on Cu foam [66].

**Table 2 materials-18-04070-t002:** Summary of latest studies using carbonate/bicarbonate feed.

Cathode Catalyst	Catholyte	FE_CO_ (%)	FE_CH4_ (%)	FE_C2H4_ (%)	CO_2_ Outlet (%)	Current (mA·cm^−2^)	Cell Voltage (V)	Reference
Cu-Sn(B)	CO_2_ (g) sat. in 0.5 M KHCO_3_ and 0.01 M EDTA	50	0	0	47	−50	−2.2	This work
Cu-Sn(A)	CO_2_ (g) sat. in 0.5 M KHCO_3_ and 0.01 M EDTA	12	3	35	35	−100	−2.7	This work
Cu-Sn(C)	CO_2_ (g) sat. in 0.5 M KHCO_3_ and 0.01 M EDTA	17	26	5	32	−100	−2.7	This work
Cu*-Sn electrodeposited	CO_2_ (g) sat. in 0.5 M KHCO_3_ and 0.01 M EDTA	62	0	1	49	−25	−2.0	[54]
Ag composite	3 M KHCO_3_	82	nr **	nr	nr	−100	−3.4	[64]
Porous Ag	3 M KHCO_3_	60	nr	nr	nr	−100	−3.7	[58]
Ag nanoparticles	2 M KHCO_3_	46	nr	nr	41	−200	−3.8	[68]
Electrodeposited Ag	2 M KHCO_3_ with 0.02 M DTAB	85	nr	nr	50	−100	−3.5	[67]
Ag foam	3 M KHCO_3_	15	nr	nr	~45	−500	−2.2	[69]
Cu foam	3 M KHCO_3_ with 3 mM CTAB	~9	27	nr	nr	−400	−7.2	[66]
Cu-Sn bronce	CO_2_ (g) with 0.1 M KHCO_3_	85	nr	~ 0	nr	−6	−0.8 V vs. RHE	[42]

** “nr” stands for “not reported”.

In this work, DTAB-modified Cu-Sn(B) catalyst emerges as a standout, delivering 50% FE_CO_ at just −2.2 V, outperforming Ag foam’s 15% FE_CO_ at the same voltage [69] and operating at significantly lower overpotentials than Ag composites (−3.4 V to −3.8 V). It reaches FE_CO_ values very close to our previous work, but at a higher current density. Critically, Cu-Sn(B) completely suppresses competing CH_4_/C_2_H_4_ formation, enabling clean syngas production (CO:H_2_ ≈ 1:1) ideal for industrial use.

## 4. Conclusions

This study demonstrates the successful development of Cu-Sn bimetallic catalysts via electrodeposition for the efficient electrochemical reduction in CO_2_ (CO2RR) to value-added products, particularly CO and syngas. The optimized Cu-Sn(4.5) catalyst, prepared at pH 4.5, achieved a remarkable Faradaic efficiency (FE) of 50% for CO at −50 mA·cm^−2^ and 120 mL·min^−1^, while the DTAB-modified Cu-Sn(B) catalyst further enhanced performance, reaching 50% FE for CO at the same current density but with a lower cell potential (−2.17 V vs. −2.46 V for Cu-Sn(4.5)), highlighting improved energy efficiency. These results are particularly promising for industrial syngas production, as the CO:H_2_ ratios obtained (e.g., surpassing or around 33:66) align with the requirements for downstream processes like Fischer-Tropsch synthesis.

Beyond CO, the catalysts also exhibited notable selectivity toward C_1_ and C_2_ products. For instance, Cu-Sn(A) achieved an FE of 35% for C_2_H_4_ at −100 mA·cm^−2^, while Cu-Sn(C) yielded 26% FE for CH_4_ under similar conditions. The synergy between Cu and Sn, along with the use of DTAB surfactant, improved catalyst homogeneity and stability, mitigating detachment during operation. Structural characterization revealed that Cu(OH)_2_ formations during CO2RR may play a role in enhancing performance, warranting further investigation.

These findings underscore the potential of electrodeposited Cu-Sn catalysts for scalable CO_2_ conversion, offering a cost-effective and tunable route to syngas and multicarbon products. Future work should focus on optimizing Sn distribution and exploring long-term stability to advance toward practical applications in renewable energy-integrated systems.

## Figures and Tables

**Figure 1 materials-18-04070-f001:**
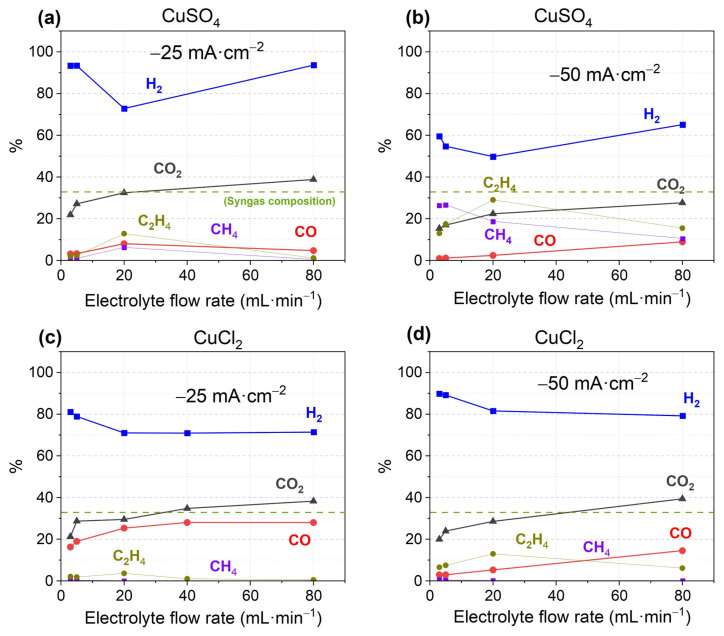
CO2RR FE performance of various products and outlet volume % of CO_2_ of (**a**,**b**) CuSO_4_ catalyst and (**c**,**d**) CuCl_2_ catalysts. For reader’s clarity, syngas composition is shown (green dashed line).

**Figure 2 materials-18-04070-f002:**
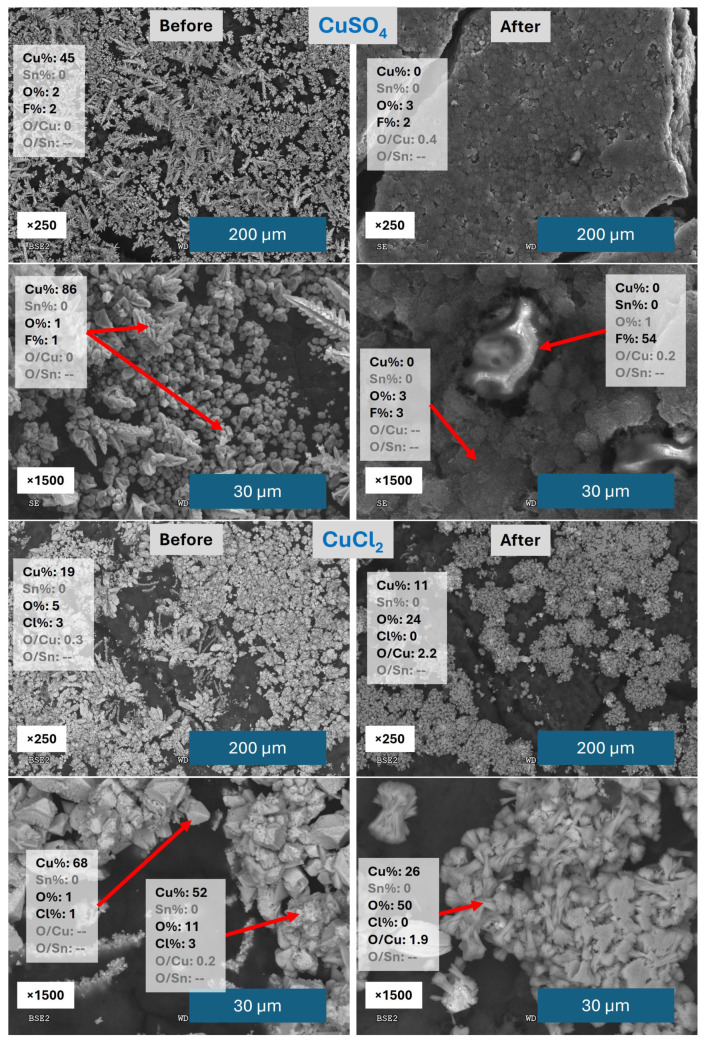
SEM/EDX images and atomic percentage results of CuSO_4_ and CuCl_2_ catalysts before and after CO2RR. The red arrows indicate the region of the image over which the atomic analysis is shown.

**Figure 3 materials-18-04070-f003:**
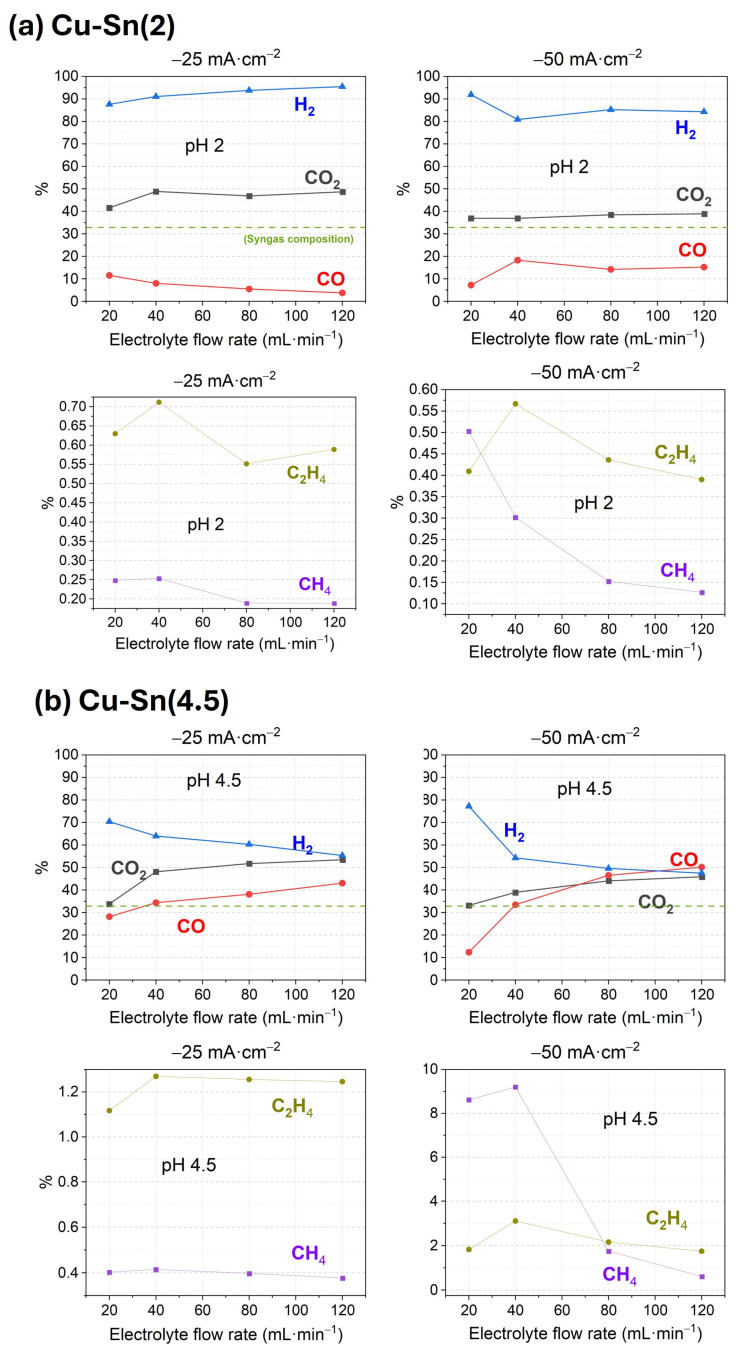
CO2RR FE performance for various products and outlet volume % of CO_2_ of (**a**) Cu-Sn(2) and (**b**) Cu-Sn(4.5) catalysts.

**Figure 4 materials-18-04070-f004:**
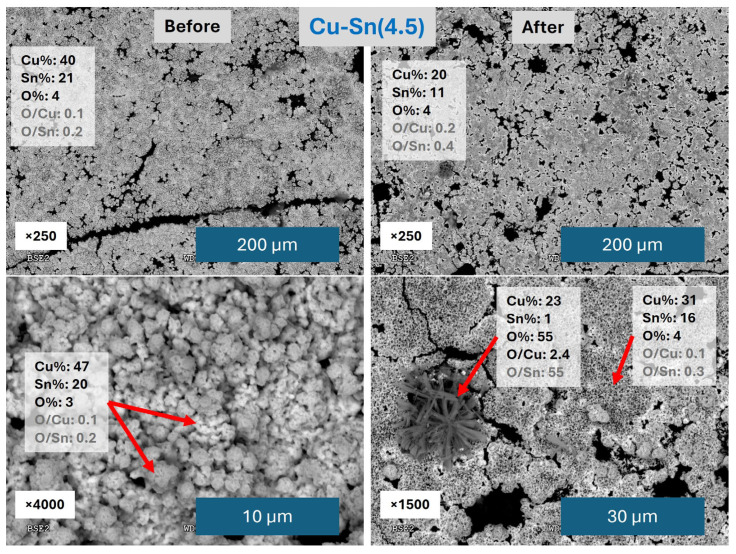
SEM/EDX images and atomic percentage results of Cu-Sn(4.5) catalyst before and after CO2RR. The red arrows indicate the region of the image over which the atomic analysis is shown.

**Figure 5 materials-18-04070-f005:**
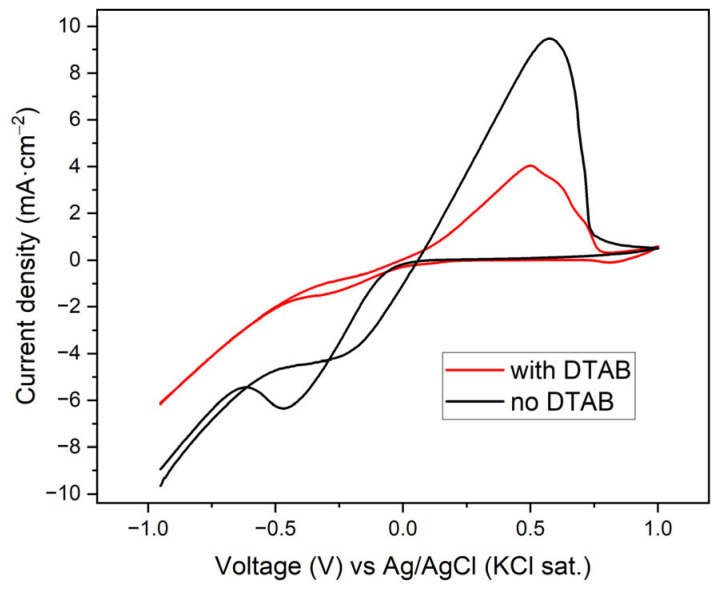
CVs at 20 mV·s^−1^ of carbon cloth in 50 mM CuSO_4_ at pH 2 with and without 5 mM DTAB.

**Figure 6 materials-18-04070-f006:**
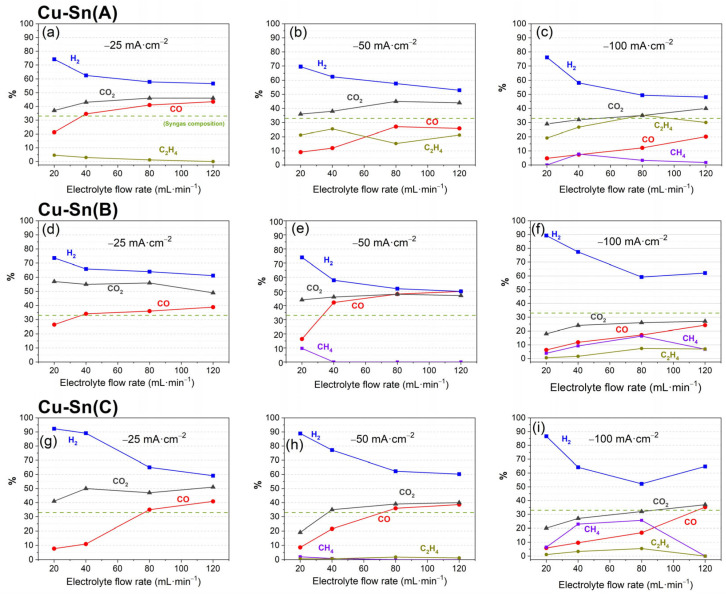
CO2RR FE performance for various products and outlet volume % of CO_2_ of (**a**–**c**) catalyst Cu-Sn(A); (**d**–**f**) catalyst Cu-Sn(B), and (**g**–**i**) catalyst Cu-Sn(C).

**Figure 7 materials-18-04070-f007:**
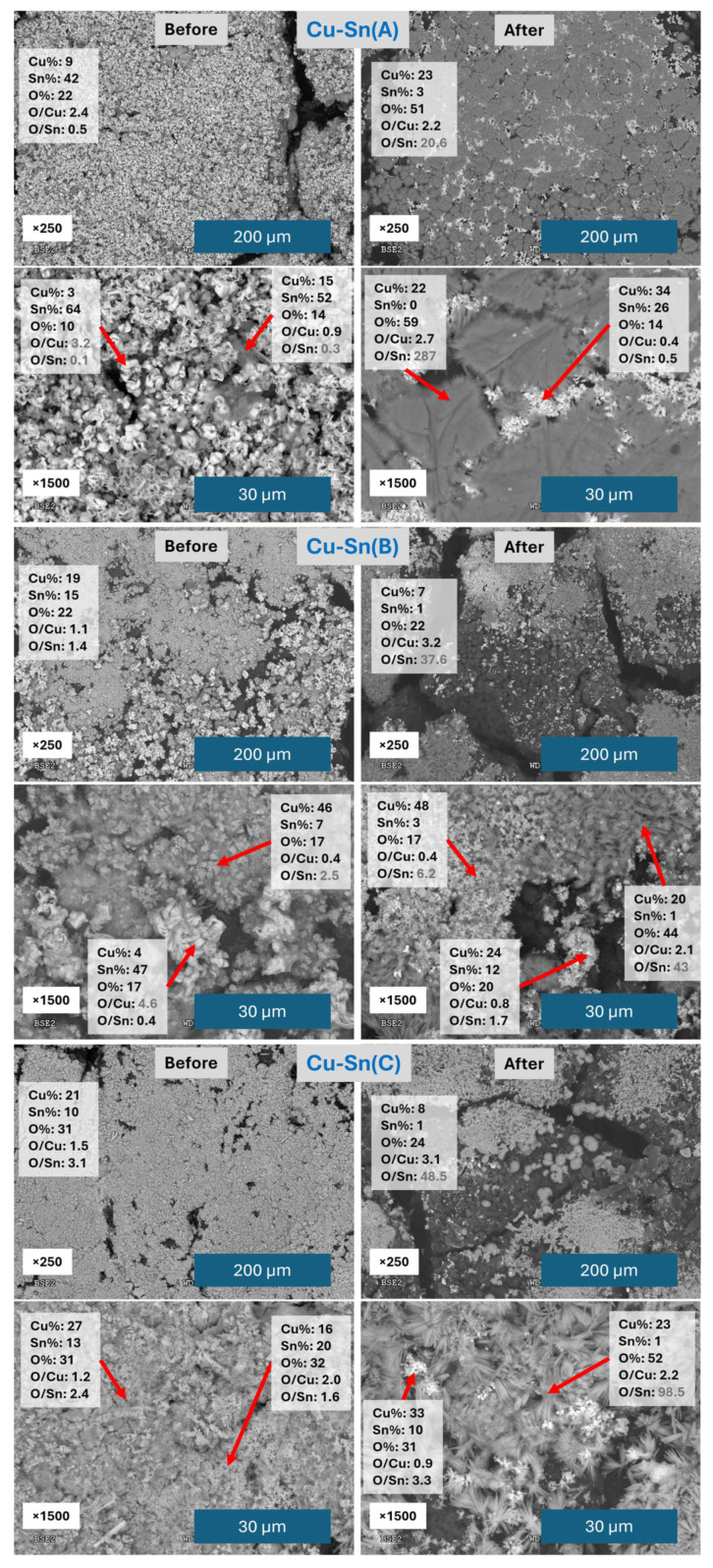
SEM/EDX images and atomic percentage results of the catalysts Cu-Sn(A), Cu-Sn(B), and Cu-Sn(C). The red arrows indicate the region of the image over which the atomic analysis is shown.

**Table 1 materials-18-04070-t001:** Bimetallic catalysts prepared by CA for CO2RR testing.

Catalyst	Solutions	Use of DTAB	N° of CAs 1200 s	Potential vs. Ag/AgCl (V)	Picture of Electrodeposited Carbon Cloth	Zoom with Optical Microscope (50×)
Cu-Sn(A)	CuSO_4_ 50 mM SnCl_2_ 50 mM	Only in CuSO_4_	CuSO_4_ (2 CAs) SnCl_2_ (1 CA)	CuSO_4_ (−0.900 V) SnCl_2_ (−0.700 V)	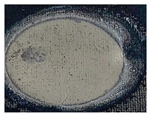	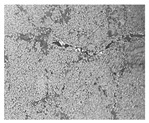
Cu-Sn(B)	CuSO_4_ 50 mM SnCl_2_ 50 mM	Both	CuSO_4_ (2 CAs) SnCl_2_ (1 CA)	CuSO_4_ (−0.500 V) SnCl_2_ (−0.600 V)	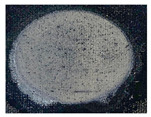	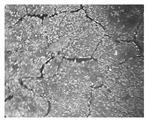
Cu-Sn(C)	CuSO_4_ 50 mM SnCl_2_ 50 mM	Both	CuSO_4_ (2 CAs) SnCl_2_ (1 CA)	CuSO_4_ (−0.600 V) SnCl_2_ (−0.650 V)	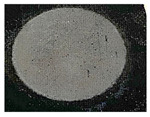	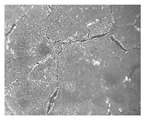

## Data Availability

The original contributions presented in this study are included in the article/Appendix A. Further inquiries can be directed to the corresponding author.

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
