# Peer review of "Modified Cu-Sn Catalysts Enhance CO2RR Towards Syngas Generation"

_materials, 2025, doi:10.3390/ma18174070_

Round 1

Reviewer 1 Report

Comments and Suggestions for Authors

In this manuscript, the authors reported how tuning the electrodeposition parameters (pH, surfactant DTAB, and metal precursors) can be used to engineer Cu-Sn catalysts for selective CO2 electroreduction for various products. Overall, this work has good novelty and the results can have strong implications for the development of electrocatalytic CO2 reduction based on Cu catalysts. I would like to recommend publication at the Materials journal. However, before possible acceptance, several specific comments given below require to be properly addressed so as to further enhance the quality and clarity of this research work.

  1. For all the CO2 electroreduction data in Figures 2, 4 and 6, for the y axis, the authors simply put “%”. It is suggested that the authors revise into “FE (%)” for clarity.
  2. For all the SEM/EDX images, Figures 3, 5, and 7, the authors quantified the amount of Cu, Sn and O. Were these percentages in atomic ratio or weight ratio? Why didn’t the percentages of the three elements add up to 100%?
  3. To appeal to a broader readership, related works on Cu-based CO2 reduction electrocatalysis can be referenced (doi: 10.1016/j.matre.2023.100175).
  4. Line 79, the authors termed the electrochemical reduction of CO2 to CO as “CO2RR”. This is not accurate because the CO2 electroreduction generates multiple types of products beyond CO, which is also the case of this research work. The authors are suggested to revise the definition of “CO2RR”.
  5. KHCO3 electrolyte is commonly used for electrochemical CO2 reduction measurement. The authors added EDTA into the electrolyte. Please justify why EDTA was added.
  6. Line 51, Equation 1, the “CO2” on the left side of the equation was mistakenly written as “O2”. The authors included both alkaline and acidic reactions for the HER and OER, however, for the CO2 reduction, only the alkaline reaction was provided.
  7. In multiple areas of the manuscript, the authors described the electrolyte as “KHCO3 saturated CO2 solution”. The authors might actually want to mean “CO2-saturated KHCO3 solution”. Please double check and revise accordingly.

Author Response

Authors: please see the attached file.

Reviewer 2 Report

Comments and Suggestions for Authors

See attached PDF.

Author Response

Authors: please see the attached file.
